# Residential Wood Combustion in Finland: PM_2.5_ Emissions and Health Impacts with and without Abatement Measures

**DOI:** 10.3390/ijerph16162920

**Published:** 2019-08-14

**Authors:** Mikko Savolahti, Heli Lehtomäki, Niko Karvosenoja, Ville-Veikko Paunu, Antti Korhonen, Jaakko Kukkonen, Kaarle Kupiainen, Leena Kangas, Ari Karppinen, Otto Hänninen

**Affiliations:** 1Finnish Environmental Institute (SYKE), Latokartanonkaari 11, 00790 Helsinki, Finland; 2National Institute for Health and Welfare (THL), 70701 Kuopio, Finland; 3Faculty of Health Sciences, School of Pharmacy, University of Eastern Finland (UEF), 70210 Kuopio, Finland; 4Finnish Meteorological Institute (FMI), 00560 Helsinki, Finland

**Keywords:** residential wood combustion, population exposure, disease burden, mortality, morbidity, particulate matter, fine particle concentrations

## Abstract

Exposure to fine particles in ambient air has been estimated to be one of the leading environmental health risks in Finland. Residential wood combustion is the largest domestic source of fine particles, and there is increasing political interest in finding feasible measures to reduce those emissions. In this paper, we present the PM_2.5_ emissions from residential wood combustion in Finland, as well as the resulting concentrations. We used population-weighed concentrations in a 250 × 250 m grid as population exposure estimates, with which we calculated the disease burden of the emissions. Compared to a projected baseline scenario, we studied the effect of chosen reduction measures in several abatement scenarios. In 2015, the resulting annual average concentrations were between 0.5 and 2 µg/m^3^ in the proximity of most cities, and disease burden attributable to residential wood combustion was estimated to be 3400 disability-adjusted life years (DALY) and 200 deaths. Disease burden decreased by 8% in the 2030 baseline scenario and by an additional 63% in the maximum feasible reduction scenario. Informational campaigns and improvement of the sauna stove stock were assessed to be the most feasible abatement measures to be implemented in national air quality policies.

## 1. Introduction

Exposure to air pollution, especially fine particles (PM_2.5_), is linked to substantial impacts on public health (e.g., [1,2]). Concentrations of air pollutants in ambient air are formed from a combination of long-range transported pollutants, as well as national and local emissions. Lehtomäki et al. [3] estimated the burden of disease and deaths in Finland attributable to ambient particulate matter (PM_2.5_ and PM_10_), nitrogen dioxide, and ozone exposures from all sources in 2015. They estimated PM_2.5_ to cause 26,000 disability-adjusted life years (DALY) and 1600 deaths, which comprised 75% and 80% of the total disease burden of air pollution, respectively.

Particulate emissions from most major sources, like transport and industrial-size combustion plants, have been regulated by law for decades. The implementation of continually stricter legislation has effectively decreased the emissions during this time [4]. Emissions from residential wood combustion (RWC), however, have not been regulated thus far. RWC has become the largest source of PM_2.5_ in many European countries [5] and is estimated to be among the most important contributors to air quality in urban areas [6]. In Helsinki, PM_2.5_ concentrations attributable to RWC have been estimated to be 18–29% in urban measurement sites and 31–66% in suburban measurement sites during the heating season [7]. RWC sources were found to be responsible for 12–40% of the local urban contributions to the PM_2.5_ concentrations at four stations in the Helsinki Metropolitan Area annually in the 2010s [8]. However, due to the substantial regional background of the PM_2.5_ concentrations, the contributions of RWC to the total PM_2.5_ concentrations in the Helsinki region ranged from negligible to approximately 15% at those stations. The relative contributions of RWC to the PM_2.5_ concentrations have also substantially increased since the 1980s [8]. In another study, it was found that the annual average contributions of RWC to the PM_2.5_ concentrations within the Helsinki region in 2013 ranged, spatially, from negligible to 15% [9]. The RWC contributions were largest within the suburban areas situated to the west, north, and northeast of the city centre of Helsinki.

Many intergovernmental treaties, such as the EU’s National Emission Ceilings (NEC) directive (2016/2284), require countries to control their emissions and achieve specific emission reduction goals. The NEC directive also compels EU countries to produce an air pollution control programme, to ensure that necessary measures will be identified and implemented to reach those goals. As RWC is becoming relatively more important as a source of air pollutants, figuring out feasible emission reduction measures and policies has recently received increasing attention.

In this work, we calculate the PM_2.5_ emissions from RWC in Finland and model the resulting concentrations in ambient air. Concentrations are used to calculate population exposure to PM_2.5_ due to RWC, and assess the resulting health impacts. We also assess the projected change in emissions between 2015 and 2030, and study a set of abatement measures for their efficiency and applicability. The objectives of this work are to: (1) Quantify the negative health impacts caused by RWC in Finland; (2) estimate how the impacts will change in the future, with and without additional abatement measures; and (3) point out feasible measures to be implemented in national air quality policies. The results were used in the planning of the National Air Pollution Control Programme 2030.

## 2. Materials and Methods

### 2.1. PM_2.5_ Emissions

In this work, RWC includes all wood burning in residential and recreational houses, as well as direct wood heating of commercial, industrial, and agricultural buildings. The emissions for base year 2015 and scenarios up to 2030 were estimated with the national emission scenario model FRES [10]. The main principles of the emission calculations in the model are explained below, and the detailed calculation scheme for RWC emissions have been reported in Savolahti et al. [11,12]. 

Emissions of PM_2.5_ are a product of wood consumption and emission factors for a given appliance. For the year 2015, the amount of wood consumption was taken from Statistics Finland [13], and for 2030 from the latest national Energy and Climate Strategy [14]. The allocation of wood use to various appliance types was based on national questionnaire surveys [15], as well as other available information, historical trends, and expert judgement. The primary energy of total wood consumption in the residential sector was 61 PJ in 2015, and it is estimated to increase to 67 PJ in 2030. Most of the wood is used for heating. The share of cooking and recreational use cannot be explicitly extracted from the surveys. Most common appliances are various kinds of masonry stoves, automatic or manual boilers and sauna stoves. Detailed wood consumption by appliance type and the projected changes in the appliance stock have been presented earlier in Savolahti et al. [12]. The calculation of emissions includes specific emission factors for five types of small-scale boilers and eight types of stoves, based mostly on Finnish measurements. It also includes separate emission factors for normal and smouldering combustion of stoves, as well as the assumed share of smouldering combustion in the baseline. Smouldering combustion represents the situation with typical user mistakes, namely suboptimal batch sizes and ignition, insufficient air supply, and poor quality of fuel (wet wood or waste). The shares of normal and smouldering combustion determine the applied emission factor of an appliance. All emission measurements have been conducted from diluted flue gases. In normal combustion, PM_2.5_ emission factors for stoves range from 33 mg/MJ (modern masonry heater) to 578 mg/MJ (open fireplace). For boilers the range is even larger; from 16 mg/MJ (wood chip boiler) to 700 mg/MJ (manually-fed log boiler without accumulator). 

Total annual emissions were spatially distributed into a 250 m × 250 m grid, using several proxies. The emissions were distributed into detached and semidetached houses based on their average wood consumption. The wood consumption of a house depended on: (1) Main heating method, (2) residential area type and (3) heating degree day. The average wood consumptions were based on questionnaires. The calculation parameters are presented in Paunu et al. [16]. Distribution of recreational building emissions was based on floor space and heating degree day. Consumption in boilers was distributed only to buildings with wood heating as the primary heating method. The national building and dwelling register was used to identify building locations and primary heating methods. For supplementary heating (use of stoves), only detached and semidetached houses were assumed to have consumption. Houses in urban areas of Southern Finland, with district heating as their primary heating method, were supposed to use the least wood for supplementary heating. In other building types than detached and semidetached houses, i.e., row and apartment houses, wood burning appliances are rare (in 2015 they attributed to less than 2% of the fuelwood consumed in residential buildings [13]), and they were not taken into account in the spatial distribution of emissions. No estimation was made for spatial changes in the building stock after 2015. Thus, the spatial distribution of emissions followed the same principles for 2015 and 2030, although total emissions and emissions per unit of fuel were assumed to change during the time span.

### 2.2. Studied Scenarios and Abatement Measures

The baseline scenario and scenarios including additional measures to reduce emissions from residential wood combustion are described in Table 1. Each studied scenario includes an additional emission reduction measure, which is assumed to be implemented cumulatively on top of the previous one(s). The order of the measures is based on the ease of implementation and/or their social acceptability; we considered scenario 1 to be the easiest to implement. This way, the scenarios represent the level of ambition in mitigation, and the last scenario (4) can be viewed as maximum feasible reduction (MFR). 

Informational campaigns for better stove use and wood storing have already been conducted in some Finnish municipalities. However, it has been difficult to evaluate the impact of such campaigns, and potential effects have been numerically explored in Savolahti et al. [11]. For sauna stoves, a harmonised emission measurement protocol will be needed for the labelling of a “modern stove”. The possibility to create and apply such a protocol is currently being explored in Finland. Small-scale electrostatic precipitators (ESPs) for residential use are available, but the technology is not currently widely used, and these will need to be further developed before a widespread implementation will be possible. Combustion bans have been opposed by most decision-makers up to date; these have therefore been considered to be the last resort measure.

### 2.3. Dispersion of Emissions

The treatment of the atmospheric dispersion processes in the study is based on the use of pre-computed source-receptor matrices (SRMs) with a resolution of 250 m × 250 m. Matrices are derived from a Gaussian dispersion model, Urban Dispersion Modelling system (UDM-FMI) [17]. In computing the SRMs, meteorological data for ten different Finnish locations were used, from a period of 2000–2005. For these ten spatial domains, the source receptor matrices were then calculated separately on a monthly level. The different temporal distributions of wood use in the appliance classes were taken into account in the monthly distribution of the emissions. For simplicity, the release heights, including the plume rise, for all the residential combustion emissions were assumed to be 7.5 m. The principles of the method are explained in Karvosenoja et al. [18], based on coarser resolution SRMs. The impact of climate change on the meteorological conditions was not taken into account in the dispersion modelling.

### 2.4. Assessment of Population Exposure

On a country level, building and dwelling register (BDR) population data for 2014 was used to calculate population-weighted primary fine particle (PPM_2.5_) concentration (PWC), originating from residential wood combustion. The same population data was used for 2030 to assess the impact of changing emissions only. Statistics Finland population grid data (Figure 1) at 1 km resolution, presented in a Finnish geographic coordinate system (ETRS-TM35FIN), were used to calculate PWC for municipalities [19].

Population-weighted concentrations were calculated according to Equation (1).
(1)$$PWC=\frac{\sum_{i=1}^{n}C_{i}P_{i}}{\sum_{i=1}^{n}P_{i}},$$
where *n* is the number of grid cells, *C* is the outdoor concentration in *i*th cell, and *P* is number of persons in *i*th cell.

### 2.5. Assessment of Health Impacts

Health impacts of RWC were estimated using the health impacts of air pollution (ISTE) model developed at the National Institute for Health and Welfare (THL) [3]. The ISTE model estimates the health impacts in DALYs, years of life lost (YLLs), years lived with disability (YLDs), and number of deaths using burden of disease methods [20]. The model includes global health estimates (GHE) baseline health data by WHO for 2015 [21], and a large set of concentration-response (C-R) functions. The C-R functions used in this work are listed in Table 2. A log-linear shape was assumed.

Baseline health burden for natural mortality was calculated by subtracting injuries and violent causes (GHE code 151) from all causes (GHE code 0). Morbidity (YLD) of cardiovascular diseases (GHE code 110) and respiratory diseases (118) were used for baseline health data instead of hospital admission data. 

The attributable disease burden (EBoD) was calculated by combining population attributable fraction (PAF) with background disease burden (BoD) (Equation (2)).
(2)EBoD=PAF×BoD
where PAF is calculated as Equation (3).
(3)$$\mathrm{PAF}=\frac{f\times \left(RR_{E}-1\right)}{f\times \left(RR_{E}-1\right)+1}$$
where f is the percentage of the exposed population in the whole target population, and *RR_E_* is the relative risk of the population at the prevailing exposure level.

Burden of disease attributable to different RWC scenarios was calculated using the attribution method [23] (Equation (4)).
(4)$$EBoD_{RWC}=\frac{PWC_{RWC\_on}-PWC_{RWC\_off}}{PWC_{RWC\_on}}\times EBoD_{tot\left(RWC\_ON\right)}$$
where $EBoD_{RWC}$ is the burden of disease attributable to the scenario, $PWC_{RWC\_on}$ is the total population-weighted PM_2.5_ concentration, $PWC_{RWC\_off}$ is the total population-weighted PM_2.5_ concentration without concentration from the scenario, and $EBoD_{tot\left(RWC\_ON\right)}$ is the total burden attributable to PM_2.5_.

Health benefits of abatement measures were estimated using the subtraction method [23] by subtracting from the total burden of disease attributable to PM_2.5_ the burden which is attributable to other sources of PM_2.5_ than RWC (Equation (5)).
(5)$$EBoD_{RWC}=EBoD_{tot\left(RWC\_ON\right)}-EBoD_{tot\left(RWC\_OFF\right)}$$
where $EBoD_{RWC}$ is the burden of disease attributable to residential wood combustion scenario which can be reduced if the scenario is implemented, $EBoD_{tot\left(RWC\_ON\right)}$ is the burden of disease attributable to total PM_2.5_ (including RWC), and $EBoD_{tot\left(RWC\_OFF\right)}$ is the burden of disease attributable to PM_2.5_ from other sources (excluding RWC). 

## 3. Results

### 3.1. Emissions in the Baseline and Reduction Scenarios

Total PM_2.5_ emissions from RWC were 10.5 Gg in 2015 (Table 3). In the baseline scenario, emissions decreased slightly from 2015 to 2030, despite a small increase in wood consumption. This is due to the Ecodesign Directive (Commission Regulations (EU) 2015/1185 and 2015/1189), and other changes in the appliance stock. The average emission factors of stove and boiler stocks are assumed to decrease slowly, as the equipment is updated. The Ecodesign Directive will further enhance this turnover rate, as all new stoves (except for sauna stoves) and boilers must meet the emission limit requirements after 2022 and 2020, respectively. Also, the share of pellets as a fuel in boilers is assumed to increase in the baseline, and the most polluting manually-fed boilers are expected to be almost phased out by 2030. The scenarios represent the cumulative impact in reduced emissions when each new measure is implemented on top of the previous one(s). The biggest reductions are achieved with the implementation of the sauna stove legislation and ESP installations to boilers. With maximum feasible reduction, the emissions would decrease by almost 50% from the 2030 baseline. Sauna stove legislation and urban combustion bans are measures that overlap with previous ones in the scenarios, which is why the impact of those measures cannot be fully singled out. If either of those measures was to be implemented separately on top of the baseline, their impact on emission reductions would be slightly larger than the differences between scenarios 1 and 2 or 3 and 4.

### 3.2. Resulting PM_2.5_ Concentrations in Ambient Air

Modelled annual averages of PM_2.5_ concentrations [µg/m^3^] resulting from RWC are presented in Figure 2. The highest concentrations (up to ~2 µg/m^3^) occur in the residential areas of the largest cities, while most areas in Southern Finland are somewhat affected. In 2015, the measured PM_2.5_ concentrations in the Helsinki Metropolitan Area were between 5 and 8 µg/m^3^ [8], and therefore the >1 µg/m^3^ around major cities is a notable increase to the background concentrations. All population centres are visible on the concentration maps, but on closer inspection, the downtown areas of major cities have lower concentrations than the surrounding suburbs. This is due to downtowns having a large number of apartment buildings, to which no wood consumption is allocated. On the other hand, the highest densities of detached buildings are in the suburbs of major cities. The modelling result is in line with measurements (see chapter 4). Only a slight decrease in the concentrations is noticeable between 2015 and the 2030 baseline.

The reductions of modelled PM_2.5_ concentrations in each scenario are presented in Figure 3. Infocampaign (that affects all stove types) and sauna legislation show effects with similar spatial patterns. As sauna stoves are widely used in all area types, where other residential wood combustion also takes place, they have not been treated separately from other supplementary heating in the spatial distribution of emissions. However, recreational and residential houses have been treated separately, and sauna stoves are relatively more common in recreational houses. Due to this, the impacts of scenario 2 were weighed slightly more towards rural areas than the impacts of scenario 1. Sauna legislation was estimated to be more effective in reducing emissions, which is why the changes in concentrations are considerably higher in scenario 2. As ESP installations only affect boilers, which are more common in rural areas, the changes in concentrations are located mostly in those areas. Effects are visible throughout Finland, but are most pronounced in Western Finland, where boilers are more common according to the national building and dwelling register. In the fourth scenario where urban combustion bans are added, the centres of the most population-dense municipalities are highlighted. While reducing the concentrations significantly in the biggest population centres, the combustion ban does not cover many of the smaller municipalities, which still have suburban areas with considerable concentrations and population exposure. 

### 3.3. Health Impacts

The burden of disease estimates for the evaluated scenarios are presented in Table 3, and the reductions of annual deaths between the scenarios in Figure 4. The estimated disease burden caused by RWC was 3420 DALY in 2015, which decreased to 1150 DALY in the MFR scenario for 2030. Deaths attributable to RWC were 204 in 2015 and 68 in the MFR scenario in 2030. Overall, the disease burden was dominated by mortality, as YLL comprised ~98% of DALY. 

The largest reductions in health impacts were achieved in scenarios 2 and 4, with the additions of sauna legislation and urban combustion bans (Figure 4). When calculated as reductions in health impacts per abated unit of emissions, installing ESPs was the least efficient measure. Measures specifically targeted to urban areas were the most efficient, as expected. Urban combustion ban was the most efficient, since it only targeted emissions in the actual population centre, whereas infocampaigns in cities were assumed to effect emissions in the whole municipality, including recreational houses.

Annual deaths in 2015 attributable to RWC, classified by population in each municipality, are shown in Table 4. The largest health impacts occur in municipalities with a population size of 20,000–50,000 inhabitants. In 2015, it was the group with second most population after the biggest cities of 200,000+ inhabitants [24]. Attributable deaths per capita in the municipality groups was highest in group of 50,000–100,000 inhabitants, although all groups between 10,000 and 200,000 inhabitants were relatively even. Incidence of deaths per capita was lowest in the group of 200,000+ inhabitants.

## 4. Discussion

Some verification to our emission and dispersion modelling can be provided by measurements made previously. Measured annual total concentrations from all sources in Finnish urban and suburban areas are typically between 5 and 8 µg/m^3^. In those areas, our modelled concentrations attributable to RWC were typically between 0.5 and 2 µg/m^3^. Our modelled PM_2.5_ concentrations, caused by RWC in 2015, were in line with measured estimations of the share of RWC contribution to total concentrations [7]. However, a direct comparison cannot be made due to differences in the study settings. 

Lehtomäki et al. [3] calculated the total burden of disease in Finland attributable to PM_2.5_ from all sources. Compared with their results, the disease burden attributable to RWC in our work would constitute approximately 13% of the total disease burden caused by PM_2.5_ concentrations in Finland. The uncertainties due to concentration-response relationships in the burden of disease estimates for different RWC scenarios range from –34% to +39%. Lehtomäki et al. [3] estimated uncertainties related to population-weighted concentrations as well, using the same emission data as in this work, but a different dispersion model. They found that for PM_2.5_ the uncertainties were ±8%. 

One limitation of this study is that we were not able to estimate the contribution of RWC to household air pollution. Burden of disease from residential wood combustion is closely related to household air pollution as the emission source is indoors. Treating of ambient and household RWC exposure as separate risk factors can lead to double counting due to their interrelated nature [25]. Therefore, our results should not be combined with burden of disease estimates of indoor RWC exposure, in order to avoid double counting. Instead, an integrated population-weighted exposure method should be applied [25].

Some recent studies suggest that the C-R curves are supra-linear at low PM_2.5_ concentration levels (e.g., [26,27,28]). Use of these curves would lead to relatively higher reduction potential as the risk curve is steeper than the log-linear used in this work. In our work, we applied C-R functions recommended by WHO HRAPIE working group [21] without a threshold.

In this study, we did not assess the impact of different RWC sources separately. Karvosenoja et al. [18] have calculated the population-weighed concentration (PWC) caused by PM_2.5_ emissions from primary wood heating (boilers), supplementary wood heating (stoves in residential houses), and recreational wood heating (heating in recreational houses). They used a similar modelling setting to ours, but in a coarser resolution of 1 × 1 km. They estimated that supplementary wood heating caused 68% and primary heating 29% of the PWC, respectively. Our study showed a congruent result, in that measures to boilers (primary heating) were less effective than measures to stoves. It is a robust supposition that similar relative shares can be applied to the health impacts of the three groups in our study. Assessment of population exposure and health impacts in 2030 did not account for changes in the population data after 2015. Statistics Finland [20] has estimated that total population will increase by ~5% between 2015 and 2030. On the other hand, the share of people living in municipalities of >200,000 inhabitants will also increase from 24% to 35%. 

Since the calculations of this study, a new, comprehensive set of measurements has been conducted for Finnish sauna stoves [29]. In our calculations, the emission factor for sauna stoves was based on a small sample size, which was considered an uncertainty. New measurements show that our emission factor is probably an overestimation, and suggested that the design of sauna stoves has improved in recent years. As the share of the improved stoves in the appliance stock increases, the average emission factor is assumed to decrease further, whereas in our baseline it stays constant. This then overestimates the emissions and the theoretical reduction potential in 2030. However, the best sauna stoves in the new set of measurements were considerably cleaner than our estimation for “modern sauna stove”, and the variation in emission factors between the stoves was significant. In this regard, it still seems a robust assumption that notable emission reduction potential from sauna stoves exists.

When assessing the real-world implementation potential of the studied measures, economical, technological, political feasibility, and social acceptability are important to take into account. The implementation costs of the measures (excluding combustion bans) have been assessed in Savolahti et al. [11]. Informational campaigns were estimated to clearly be the most cost-efficient measure to reduce emissions. The second most cost-efficient measure was updating the stock of sauna stoves, and installing ESPs was the least cost-efficient. In most cases, the costs of implementation are paid by the users of wood combustion appliances. In the case of informational campaigns, however, the costs are paid by the municipality or government, and the user can actually gain direct financial benefits. One problem with combustion bans is that for many people, access to wood is free, which is not the case for any alternative source of energy. From a technological point of view, small-scale ESPs are not widely available, and there are still some technological challenges to solve in order for them to be attractive for consumers. Sauna stoves with lower emissions have been developed and brought into the market by sauna stove manufacturers in earlier years, i.e., there are no technological barriers for them. However, because of the higher price and no incentives, they have not succeeded in the market, and currently there are no sauna stove models that are marketed as low-emission alternatives. Recently, Finnish sauna stove manufacturers have expressed interest and a positive attitude to the idea of bringing cleaner stoves to the market. However, a uniform, accessible, and reliable method to measure the emissions is currently lacking. Such a method needs to be established before incentives can be created to promote cleaner models.

Assessing the impact of informational campaigns involves large uncertainties. It is difficult to point out and quantify the changes in behaviour they might produce, and we have assumed the change to be substantial. On the other hand, our calculation method only takes into account the change in emissions, when the same amount of wood is combusted better. Although this can make a significant difference if the initial combustion practices are poor, the campaigns might also have other effects: It could reduce wood consumption. Increasing awareness of the harmful health impacts could lead to less unnecessary combustion in urban areas. Better understanding of the proper use of the stove might also lead to notable increases in net heating efficiency, which would then reduce the amount of wood needed to heat the house. These effects are not included in the assessment of informational campaigns. Overall, even with large uncertainties in the effect of such campaigns, their potential benefits seem to be worth the relatively low costs.

The studied measures were taken into consideration in the preparation of the National Air Pollution Control Programme 2030 [30], which aimed at giving concrete recommendations to improve air quality and reduce harmful health impacts in Finland. The preparation process included the participation of main interest groups, e.g., industries and NGOs, and public hearings. Thus, this process could be considered as a test for political and social feasibility. The measures of the first two scenarios, i.e., wide implementation of information campaigns and support for processes described above, that could lead to the introduction of low-emission sauna stoves into the market, were recommended in the Programme. The use of small-scale ESPs and combustion bans were not considered plausible measures due to technological risks and lack of social acceptance, respectively.

## 5. Conclusions

In this study, we modelled concentrations of PM_2.5_ resulting from primary particle emissions of RWC in Finland. Concentrations were calculated for 2015 and as projections for 2030, including abatement scenarios of increasing ambition level. Without measures, the resulting annual average concentrations were between 0.5 and 2 µg/m^3^ in the proximity of most cities. Disease burden attributable to RWC was estimated to be 3400 DALY and 200 deaths in 2015. This disease burden accounted for 13% of that caused by total PM_2.5_ concentrations in Finland. Most of the population exposure and thus estimated heath impacts were caused by supplementary heating in residential houses, i.e., the use of stoves. Use of boilers for primary heating, which usually happens in more rural areas, was less harmful per unit of emissions, and emissions from recreational houses were the least harmful. This perception is important when planning measures to reduce the disease burden. In the baseline scenario for 2030, attributable disease burden and deaths due to RWC emissions decreased by 8%, compared to 2015. In the MFR scenario, the decrease was 63%, compared to the 2030 baseline. The most effective measures were urban combustion bans and implementing a legislation that sets emission limits for sauna stoves. Informational campaigns targeted to urban areas were shown to be very efficient in terms of reduced disease burden per reduced unit of emissions. 

This work supports the Finnish policy-making processes as it influenced the planning of National Air Pollution Control Programme 2030. Informational campaigns and improvement of the sauna stove stock were the two measures included in the recommendations for actions given in the Programme, as means to further improve air quality in Finland. The Finnish Ministry of the Environments is currently surveying the possibilities to create a harmonised emission measurement standard for sauna stoves, upon which a policy to promote cleaner stoves could be built. Informational campaigns have mostly been carried out in the Helsinki metropolitan area and other major population centres. In this study, however, we propose that a major share of the harmful health impacts occur in relatively small municipalities of 20,000–100,000 people. Thus, successful policies to reduce the health impacts of RWC should include measures that also target these smaller municipalities.

## Figures and Tables

**Figure 1 ijerph-16-02920-f001:**
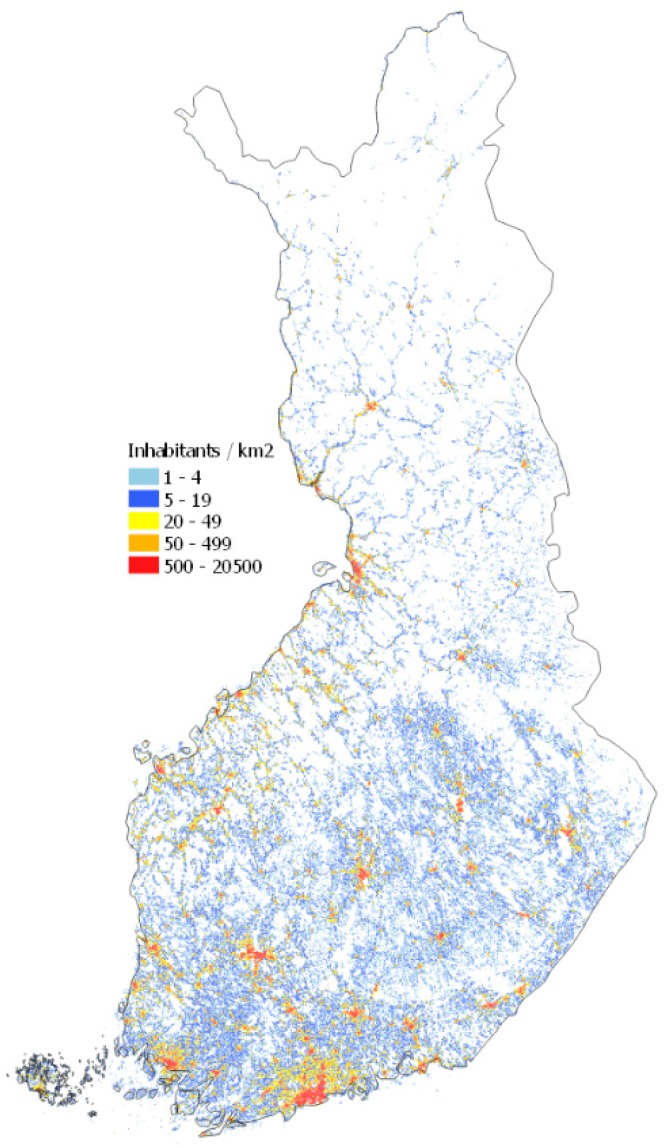
Population density in Finland at 1 km resolution.

**Figure 2 ijerph-16-02920-f002:**
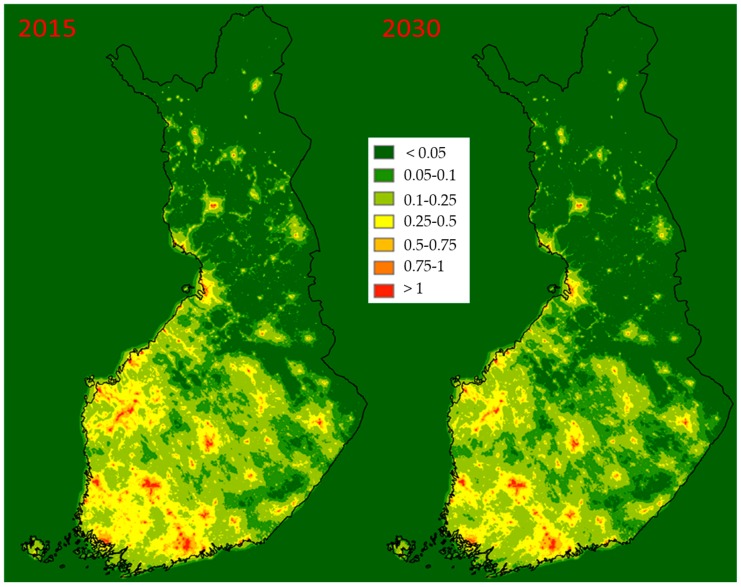
The PM_2.5_ concentrations [µg/m^3^] resulting from RWC in 2015 and the baseline scenario in 2030.

**Figure 3 ijerph-16-02920-f003:**
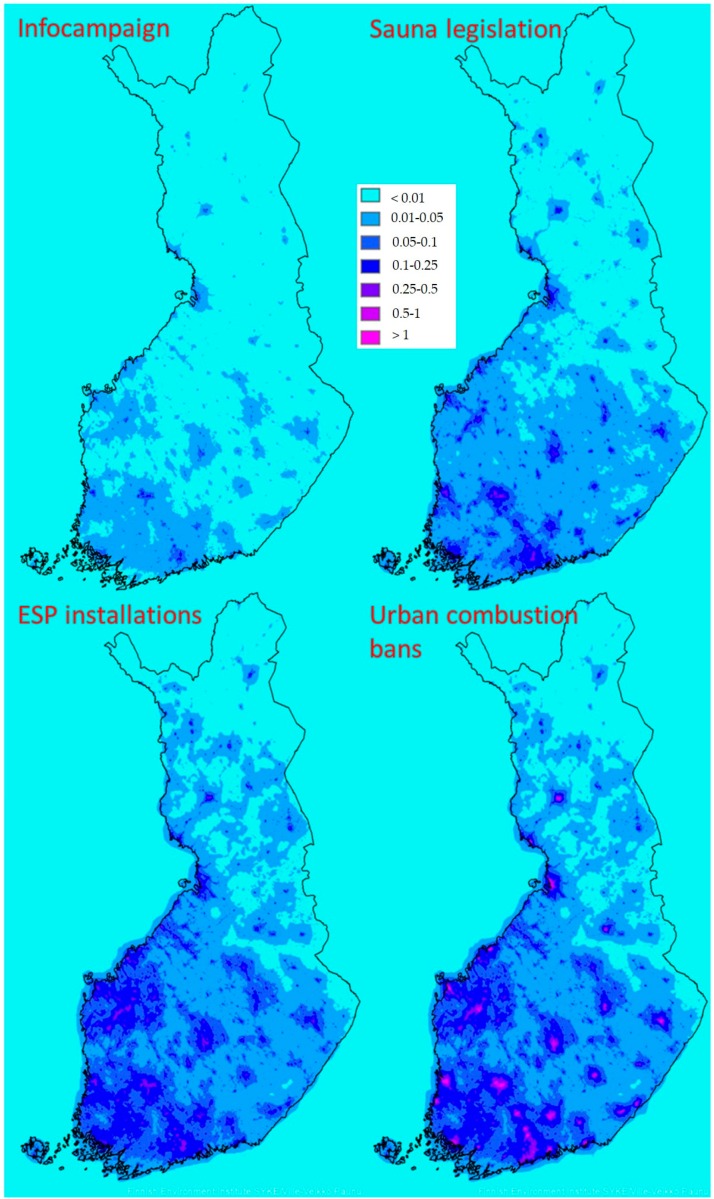
Reduction potentials of PM_2.5_ concentrations [µg/m^3^] associated with the studied scenarios.

**Figure 4 ijerph-16-02920-f004:**
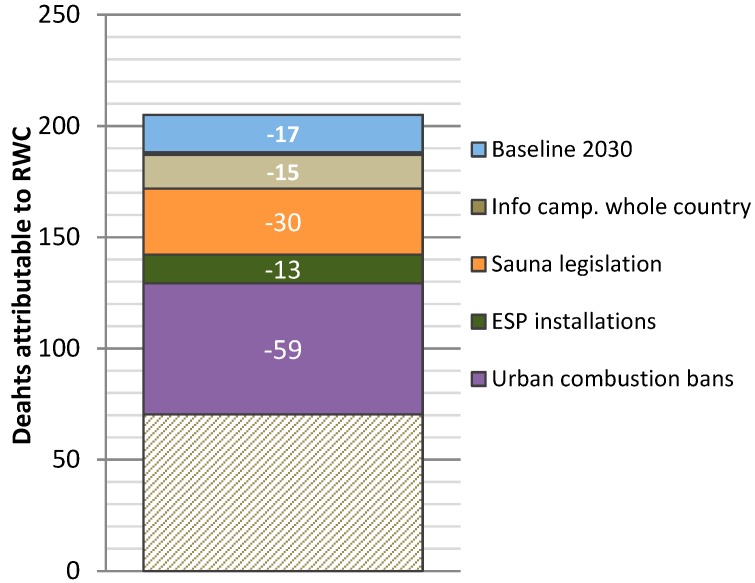
Reduction of annual deaths attributable to RWC in each scenario in 2030, with respect to the previous one, as new measures are added. Baseline 2030 has been compared to annual deaths in 2015.

**Table 1 ijerph-16-02920-t001:** Residential wood combustion (RWC) emission reduction measures in the studied scenarios. The scenarios from 1 to 4 have been applied cumulatively.

Scenarios		Description and Scope
	Baseline	Projected RWC emissions in 2030 with existing measures. Includes the Ecodesign Directive (Directive 2009/125/EC).
1	Information campaign	Training stove users in good practices of a stove. We assumed the maximum estimated effect from Savolahti et al. (2016) for the campaign. This halves the amount of smouldering combustion initially assumed for stoves.
a	Cities only	Implemented in municipalities with a population centre of 20,000+ inhabitants.
b	Whole country	Implemented throughout the country.
2	1b + national sauna legislation	On top of 1b, national regulation similar to Ecodesign, but covering only sauna stoves (which are excluded in Ecodesign): Only modern sauna stoves allowed to be sold after 2022. We assume modern sauna stoves to produce 50% fewer emissions than conventional ones. Average lifetime of a sauna stove was assumed to be 12.5 a, which was used to estimate the penetration rate of new appliances to the stock.
3	2 + ESP installations	On top of 2, requirement to install end-of-stack electrostatic precipitators (ESP) to residential-size wood boilers. Implemented to all boilers in the country. ESPs were assumed to have PM_2.5_ emission reduction efficiency of 80%.
4	3 + urban combustion ban	On top of 3, prohibiting of all residential wood combustion in urban areas of municipalities with a population centre of 20,000+ inhabitants. Urban areas were classified as grid cells with at least 200 residents and the distance between buildings less than 200 m.

**Table 2 ijerph-16-02920-t002:** Relative risk (RR) estimates with confidence intervals used in this work [22].

Health Outcome	Age Group	RR per 10 µg/m^3^ (95% CI)
Natural mortality	>30 year	1.062 (1.040–1.083)
ardiovascular diseases (hospital admissions)	All	1.0091 (1.0017–1.0166)
Respiratory (hospital admissions)	All	1.0190 (0.9982–1.0402)

**Table 3 ijerph-16-02920-t003:** Estimates for disease burden attributable to RWC in the studied scenarios. The scenarios have been evaluated with respect to Baseline in 2030. The scenarios are cumulative, i.e., they also include the measures defined in the previous scenarios. The notations: DALY = disability adjusted life years, YLL = years of life lost, YLD = years lived with disability, Δ DALY/Δ Gg PM_2.5_ = changes compared to 2030 baseline.

Scenario	PM_2.5_ Gg/a	PWC µg/m^3^	DALY	YLL	YLD	Deaths	ΔDALY/ΔGg PM_2.5_
2015	10.5	0.70	3410	3410	55	204	-
2030 Baseline	9.1	0.64	3120	3070	51	187	-
1. Infocampaign							
a Cities	9.0	0.62	2990	2940	48	179	1140
b All areas	8.5	0.59	2860	2810	46	172	400
2. Sauna legislation	7.0	0.49	2360	2320	38	141	351
3. ESP installations	5.5	0.44	2140	2100	35	128	140
4. Urban combustion bans	4.8	0.24	1140	1120	18	68	1480

**Table 4 ijerph-16-02920-t004:** Deaths attributable to RWC, classified by population size in a municipality.

Population	Inhabitants	Deaths	Deaths/100,000 Inhabitants
>200,000	1,335,224	31	2.3
100,000–200,000	737,646	30	4.1
50,000–100,000	742,217	36	4.9
20,000–50,000	1,057,312	45	4.3
10,000–20,000	686,257	28	4.1
5000–10,000	570,376	19	3.3
2500–5000	248,826	6.0	2.4
<2500	112,387	2.6	2.8

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
