# Peer review of "Residential Wood Combustion in Finland: PM2.5 Emissions and Health Impacts with and without Abatement Measures"

_ijerph, 2019, doi:10.3390/ijerph16162920_

Round 1

Reviewer 1 Report

Review
International Journal of Environmental Research and Public Health

Title
Health impacts of PM2.5 emissions from residential wood combustion in Finland, with and without abatement measures

Authors
Mikko Savolahti, Niko Karvosenoja, Ville-Veikko Paunu, Heli Lehtomäki, Antti Korhonen, Jaakko Kukkonen, Kaarle Kupiainen, Leena Kangas, Otto Hänninen, Ari Karppinen

General comments
This paper provides a health impact assessment of ambient fine particulate matter (PM2.5) exposure from residential wood combustion (RWC) in Finland. Multiple abatement scenarios of RWC emissions in 2030 are compared, relative to 2015. The study uses high–resolution dispersion modelling to estimate the impacts of changes in ambient PM2.5 concentrations on total loss of healthy life. Solid fuel use is an important and often poorly controlled source of air pollution. The topic of this paper is relevant to the scope of the International Journal of Environmental Research and Public Health. The paper is well written with interesting findings on the important contribution of RWC emissions to air pollution and disease burden in Finland. Implications for local policy are discussed, and the relatively large improvements in air quality and public health have potential for other similar countries using solid fuels.

My main concerns are the lack of justification of the health impact assessment methodology, their implications, and the alternative approaches. The current best–practise for health impact assessments from chronic ambient PM2.5 exposure are to use either the Global Exposure Mortality Model (Burnett et al 2018) or the integrated–exposure response (IER) function (Burnett et al 2014) using updated coefficients with each iteration of the Global Burden of Diseases, Injuries, and Risk Factors Study (GBD) (GBD 2017 Risk Factor Collaborators 2018). The IER and GEMM exposure–response functions have theoretical minimum risk–exposure levels (TMREL) of 2.4–5.9 g m–3, due to lack of knowledge about the shape of the concentration–mortality association at lower levels. In this study, the authors refer to Lehtomäki et al (2018) for explanation, wherein the authors mention that the TMREL of the IER/GEMM is above the PM2.5 concentrations from RWC (0.5–2 g m–3). If this is the justification for using the log–linear function from Héroux et al (2015), then this explanation would be useful for the reader given the lack of epidemiological data below 2.4 g m–3.
The IER and GEMM exposure–response functions are both non–linear, where they are supralinear at lower PM2.5 concentrations. The authors follow an attribution method by scaling the baseline disease burden by the fractional change in PM2.5 concentration. If the authors wanted to estimate the potential avoided disease burden from a change in PM2.5 concentration, then a subtraction method would be required, which subtracts the disease burden under the scenario away from the disease burden under the baseline. The difference between these approaches can be large due to the non–linear exposure–response functions. Section 2.2 of Kodros et al (2016) describes the attribution and subtraction methods.
This paper ignores the contribution of RWC to household PM2.5 pollution. When estimating the disease burden from ambient PM2.5 pollution for areas that have high residential solid fuel use, it is important to also consider household PM2.5 pollution to account for double–counting.

There are multiple methods to combine the disease burden from ambient and household PM2.5 pollution, such as in the GBD2017 (see supplementary of GBD 2017 Risk Factor Collaborators 2018) and the Integrated Population-Weighted Exposure (IPWE) (Aunan et al 2018).
Overall, this well–written paper provides important insight into a key source of air pollution, with policy–relevant approaches to improving public health. The clarity of the paper would be improved by adding the justification of the health impact assessment methodology, the implications of this choice, and the sensitivity of disease burden estimates to alternative approaches.

Specific comments
1. The acronyms NEC (line 58), UDM-FMI (line 133), ETRS-TM35FIN (line 147) are not defined upon first use.
2. The health impact assessment is central to this paper and the authors refer the reader to Lehtomäki et al (2018) to explain the bulk of the methodology. More detail on the health impact assessment would be useful for the reader, for example, how the disease burden is calculated, the exposure–response function used, and what the Health impacts of air pollution (ISTE) model is?
3. The layout of Figure 4 could cause confusion (if the reader does not carefully read the caption) and show that more stringent measures have reduced effect. The figure clarity would be improved by showing the reduction of each scenario relative to 2015 (i.e. as cumulative totals), rather than the reduction relative to the previous scenario.
4. Line 241: Should this read 200k+, rather than 200+ inhabitants?

References
Aunan K, Ma Q, Lund M T and Wang S 2018 Population-weighted exposure to PM2.5 pollution in China: An integrated approach Environ. Int. 120 111–20
Burnett R, Chen H, Szyszkowicz M, Fann N, Hubbell B, Pope C A, Apte J S, Brauer M, Cohen A, Weichenthal S, Coggins J, Di Q, Brunekreef B, Frostad J, Lim S S, Kan H, Walker K D, Thurston G D, Hayes R B, Lim C C, Turner M C, Jerrett M, Krewski D, Gapstur S M, Diver W R, Ostro B, Goldberg D, Crouse D L, Martin R V., Peters P, Pinault L, Tjepkema M, van Donkelaar A, Villeneuve P J, Miller A B, Yin P, Zhou M, Wang L, Janssen N A H, Marra M, Atkinson R W, Tsang H, Quoc Thach T, Cannon J B, Allen R T, Hart J E, Laden F, Cesaroni G, Forastiere F, Weinmayr G, Jaensch A, Nagel G, Concin H and Spadaro J V. 2018 Global estimates of mortality associated with long-term exposure to outdoor fine particulate matter Proc. Natl. Acad. Sci. 115 9592–7
Burnett R T, Arden Pope C, Ezzati M, Olives C, Lim S S, Mehta S, Shin H H, Singh G, Hubbell B, Brauer M, Ross Anderson H, Smith K R, Balmes J R, Bruce N G, Kan H, Laden F, Prüss-Ustün A, Turner M C, Gapstur S M, Diver W R and Cohen A 2014 An integrated risk function for estimating the global burden of disease attributable to ambient fine particulate matter exposure Environ. Health Perspect. 122 397–403
GBD 2017 Risk Factor Collaborators 2018 Global, regional, and national comparative risk assessment of 84 behavioural, environmental and occupational, and metabolic risks or clusters of risks for 195 countries and territories, 1990–2017: a systematic analysis for the Global Burden of Disease Stu Lancet 392 1923–94
Héroux M E, Anderson H R, Atkinson R, Brunekreef B, Cohen A, Forastiere F, Hurley F,

Katsouyanni K, Krewski D, Krzyzanowski M, Künzli N, Mills I, Querol X, Ostro B and Walton H 2015 Quantifying the health impacts of ambient air pollutants: recommendations of a WHO/Europe project Int. J. Public Health 60 619–27
Kodros J K, Wiedinmyer C, Ford B, Cucinotta R, Gan R, Magzamen S and Pierce J R 2016 Global burden of mortalities due to chronic exposure to ambient PM2.5 from open combustion of domestic waste Environ. Res. Lett. 11 1–9
Lehtomäki H, Korhonen A, Asikainen A, Karvosenoja N, Kupiainen K, Paunu V V, Savolahti M, Sofiev M, Palamarchuk Y, Karppinen A, Kukkonen J and Hänninen O 2018 Health impacts of ambient air pollution in Finland Int. J. Environ. Res. Public Health 15

Author Response

General comments
This paper provides a health impact assessment of ambient fine particulate matter (PM2.5) exposure from residential wood combustion (RWC) in Finland. Multiple abatement scenarios of RWC emissions in 2030 are compared, relative to 2015. The study uses high–resolution dispersion modelling to estimate the impacts of changes in ambient PM2.5 concentrations on total loss of healthy life. Solid fuel use is an important and often poorly controlled source of air pollution. The topic of this paper is relevant to the scope of the International Journal of Environmental Research and Public Health. The paper is well written with interesting findings on the important contribution of RWC emissions to air pollution and disease burden in Finland. Implications for local policy are discussed, and the relatively large improvements in air quality and public health have potential for other similar countries using solid fuels.

My main concerns are the lack of justification of the health impact assessment methodology, their implications, and the alternative approaches. The current best–practise for health impact assessments from chronic ambient PM2.5 exposure are to use either the Global Exposure Mortality Model (Burnett et al 2018) or the integrated–exposure response (IER) function (Burnett et al 2014) using updated coefficients with each iteration of the Global Burden of Diseases, Injuries, and Risk Factors Study (GBD) (GBD 2017 Risk Factor Collaborators 2018). The IER and GEMM exposure–response functions have theoretical minimum risk–exposure levels (TMREL) of 2.4–5.9 mg m–3, due to lack of knowledge about the shape of the concentration–mortality association at lower levels. In this study, the authors refer to Lehtomäki et al (2018) for explanation, wherein the authors mention that the TMREL of the IER/GEMM is above the PM2.5 concentrations from RWC (0.5–2 mg m–3). If this is the justification for using the log–linear function from Héroux et al (2015), then this explanation would be useful for the reader given the lack of epidemiological data below 2.4 mg m–3.

Authors’response:

Thank you for your important comments related to the health impact assessment. The discussion about the shape of the concentration-response function is very relevant.

We agree that in the global assessments the latest IER and GEMM functions are the best evidence and should be applied. In global assessments the “traditional” linear and log-linear approaches are not the best fit as there are areas with high exposures for which the (log-)linear approaches would give unrealistically high impact estimates. However, in low exposure countries the more recent supra-linear functions can be problematic due to the theoretical threshold values applied (TMREL).

The theoretical threshold values are one of the main reasons why we have not applied the named functions in this work. A recent review on health effect of PM2.5 at low exposure levels by Papadogeorgou et al. (2019) conclude the evidence to date strongly suggest no biological threshold for PM2.5 exposure and adverse health effects. IER and GEM functions though have theoretical thresholds.

Like you mentioned, IER and GEMM curves are supralinear at low exposure levels. Vodonos et al. (2018) and Hanigan et al. (2019) also suggest that the risk increase at low exposure levels is steeper than at higher exposures. The IER curves starting at zero additional risk at 2.4 µg/m3 and then rapidly increasing does not seem biologically plausible. Probably more biologically plausible shape would be for instance S-curve. Introducing a threshold due to lack of data of the shape of the curve below TMREL is not very valid argument as lack of proof does not equate a proof of lack (Levy et al. 2016).

In countries where the PM2.5 concentrations are relatively low and close to the threshold values, the use of theoretical threshold is a very important question as it can drastically reduce the estimated impacts. This can also lead to possibly false interpretation that there are only minor impacts attributable to PM2.5 in low exposure countries.

Concentrations-response functions are also under active research and for instance IER functions have already had several updates in the past few years which have big impact on the health burden in low exposure countries like Finland. It is positive that the threshold has been decreasing. We find more traditional approach suggested by WHO HRAPIE group more stable and reasonable at low exposures. In countries where the exposure levels are clearly higher than the threshold value, the threshold has relatively smaller impact on the result.

The IER and GEMM models would work better in Finland as well if we would calculate the impacts on a finer scale. In this work we calculate the health impacts in country level, this leads to some averaging in the exposure and therefore lowering the exposures in the highest exposure sells.

One challenge we have faced with the latest IER curves is the availability of the relative risk estimates for different levels of exposure. For Burnett et al 2014 the relative risk for different concentration levels were reported, but we have not seen a similar effort done for the more recent IER functions.

The IER and GEMM exposure–response functions are both non–linear, where they are supralinear at lower PM2.5 concentrations. The authors follow an attribution method by scaling the baseline disease burden by the fractional change in PM2.5 concentration. If the authors wanted to estimate the potential avoided disease burden from a change in PM2.5 concentration, then a subtraction method would be required, which subtracts the disease burden under the scenario away from the disease burden under the baseline. The difference between these approaches can be large due to the non–linear exposure–response functions. Section 2.2 of Kodros et al (2016) describes the attribution and subtraction methods.

Authors’response:

Thank you for introducing the Kodros et al. 2016 article. We changed the calculation method for reduction potential of the scenarios to the suggested subtraction method. We observed small differences in the averted burden and updated the results accordingly.

This paper ignores the contribution of RWC to household PM2.5 pollution. When estimating the disease burden from ambient PM2.5 pollution for areas that have high residential solid fuel use, it is important to also consider household PM2.5 pollution to account for double–counting.

There are multiple methods to combine the disease burden from ambient and household PM2.5 pollution, such as in the GBD2017 (see supplementary of GBD 2017 Risk Factor Collaborators 2018) and the Integrated Population-Weighted Exposure (IPWE) (Aunan et al 2018).

Authors’response:

Thank you for pointing out the importance of taking into account the contribution of household air pollution and providing references on this matter. Unfortunately, we do not have the data for indoor concentrations or time-activity patterns available for this study. We have added a note of possible double counting to discussion (lines 354-360) and aim to take the household air pollution into account in our next studies.

Overall, this well–written paper provides important insight into a key source of air pollution, with policy–relevant approaches to improving public health. The clarity of the paper would be improved by adding the justification of the health impact assessment methodology, the implications of this choice, and the sensitivity of disease burden estimates to alternative approaches.

Authors’response:

Thank you for your careful review of our manuscript and useful comments which helped us to improve the manuscript significantly. Your specific comments are addressed below.

Specific comments
1. The acronyms NEC (line 58), UDM-FMI (line 133), ETRS-TM35FIN (line 147) are not defined upon first use.

Authors’response:

Thank you for pointing this out. Definitions have been added.

2. The health impact assessment is central to this paper and the authors refer the reader to Lehtomäki et al (2018) to explain the bulk of the methodology. More detail on the health impact assessment would be useful for the reader, for example, how the disease burden is calculated, the exposure–response function used, and what the Health impacts of air pollution (ISTE) model is?

Authors’response:

We separated the assessment of health impacts in the method section to its own chapter (2.5). In this chapter we have now made the description of the methods more detailed and less dependent on the Lehtomäki et al. 2018 paper.

3. The layout of Figure 4 could cause confusion (if the reader does not carefully read the caption) and show that more stringent measures have reduced effect. The figure clarity would be improved by showing the reduction of each scenario relative to 2015 (i.e. as cumulative totals), rather than the reduction relative to the previous scenario.

Authors’ response:

Thank you for your comment, the figure is changed to one that is easier to read.

4. Line 241: Should this read 200k+, rather than 200+ inhabitants?

Authors’ response:

Yes it should be 200k+, thank you for your careful reading! Correction is made to the text.

Reviewer 2 Report

This is a very interesting manuscript dealing with reduction scenarios of PM2.5 by RWC. In general the information was well explained, nevertheless there are some issues that could be clarified.

I could not consult reference 12, but it seems that the estimations of RWC impacts in 2015 have been published before (Fig 2). Then, part of this manuscript has been accepted for publication. Please clarify this situation.

Please describe briefly the model used for estimation of RWC emissions and ISTE models as well as how they were calibrated, the used models are the center for this paper, since it is not enough to mention previous references.

The most important reduction of PM2.5 was achieved by urban combustion ban, but this urban combustion was not described. Authors used the emission inventory but for the readers the information is very scarce. The use of RWC can be very different in each country. Please in the discussion section describe which practices would be banned in the cities (chimneys, cooking, and so on),

Author Response

This is a very interesting manuscript dealing with reduction scenarios of PM2.5 by RWC. In general the information was well explained, nevertheless there are some issues that could be clarified.

I could not consult reference 12, but it seems that the estimations of RWC impacts in 2015 have been published before (Fig 2). Then, part of this manuscript has been accepted for publication. Please clarify this situation.

Authors’ response:

Thank you for pointing out that there was a word missing from the title of that article in the reference list! The paper in question addresses climate impacts of RWC. Reference list has been corrected.

Please describe briefly the model used for estimation of RWC emissions and ISTE models as well as how they were calibrated, the used models are the center for this paper, since it is not enough to mention previous references.

Authors’ response:

More text added to both models. The calibration of emission calculations can mainly be done by comparing modelled and measured concentrations, for which this manuscript is one example.

The most important reduction of PM2.5 was achieved by urban combustion ban, but this urban combustion was not described. Authors used the emission inventory but for the readers the information is very scarce. The use of RWC can be very different in each country. Please in the discussion section describe which practices would be banned in the cities (chimneys, cooking, and so on),

Authors’ response:

Explanation added to methods chapter that RWC in Finland is mostly heating, but includes cooking and recreational use. Table 1 includes the definition of urban areas. Note added to the Table that the ban includes all forms of RWC.

Reviewer 3 Report

The original article by Savolahti et al. describes the health impact of mitigating PM2.5 emission in Finland, by limiting the residential wood combustion, based on four different scenarios.

The paper is properly written, the methods clearly described, and it can be accepted for publication after minor revisions.

Few points have to be addressed:

40 - How this data compare to other European Countries?

83,164 - It is advisable to use the scientific annotation instead of unit prefix

Table 1 - How the electrostatic precipitator efficiency was chosen? 80% seems to be a cautious choice

Fig.2 - Why the concentration scale ranges between 0.05 and 1 ugm? are significative so small concentrations? A continuous instead of the discrete scale would be more appropriate for the plot. Please add units in the plot.

A discussion about the different cost (without going into details) of the different scenario would be appreciated. In particular, to show how the implementation costs would affect directly the population, rather than the Municipalities or the Government.

Author Response

The original article by Savolahti et al. describes the health impact of mitigating PM2.5 emission in Finland, by limiting the residential wood combustion, based on four different scenarios. 

The paper is properly written, the methods clearly described, and it can be accepted for publication after minor revisions. 

Few points have to be addressed:

40 - How this data compare to other European Countries?

Authors’ response:

Thank you for your question. Lehtomäki et al. 2018 focused to Finland in their study, and we did not find a natural place in the intro for a more extensive comparison. In EEA 2018 report (https://www.eea.europa.eu/publications/air-quality-in-europe-2018), similar estimates for the number of deaths related to PM2.5 in 2015 for Finland were reported. They estimated the premature deaths attributable to PM2.5 in other European countries as well. Straight comparison of the premature deaths between countries is not reasonable as the population sizes vary substantially.

Looking at the annual population weighted PM2.5 concentrations, the lowest ones were in the Nordic countries: Finland (5.3 µg/m3), Norway (5.9 µg/m3), Iceland (5.5 µg/m3), and Sweden (5.9 µg/m3). In these countries also the burden of disease attributable to PM2.5 is the lowest relatively speaking. While the highest concentrations (> 20 µg/m3) were found in Bulgaria, Poland, Albania, Former Yugoslav Republic of Macedonia. Kosovo and Serbia.

83,164 - It is advisable to use the scientific annotation instead of unit prefix

Authors’ response:

We changed kt to Gg. However, we prefer to use J instead of Wh, as it is in the International System of Units, and for the sake of consistency with our previous work.

Table 1 - How the electrostatic precipitator efficiency was chosen? 80% seems to be a cautious choice

Authors’ response:

Efficiency is based on multiple measures using an ESP in small-scale appliances (Lenz, V., Thr€an, D., Hartmann, H., Turowski, P., Ellner-Schuberth, F., Gerth, J., 2008. DBFZ Report Nr.1: Bewertung und Minderung von Feinstaubemissionen aus häuslichen Holzfeuerungsanglangen. Deutsches BiomasseForschungsZentrtum). It is true that in some cases the efficiency is considerably higher. In general however, efficiencies for ESPs tend to decrease when there is significant amount of incomplete combustion, as is often the case with small-scale wood burning appliances. For this reason reduction efficiencies are lower than what they would be in industrial-sized combustion plants.

Fig.2 - Why the concentration scale ranges between 0.05 and 1 ugm? are significative so small concentrations? A continuous instead of the discrete scale would be more appropriate for the plot. Please add units in the plot.

Authors’ response:

The steps in the scale were originally chosen to basically show the areas where RWC is occurring, in as much detail and clarity as possible. It is true that concentrations of 0,05 µg/m3 are probably not of much significance, and maybe some other scale could be more appropriate. The plots were created in one of the projects behind the paper, and unfortunately new plots could not be produced within the schedule of this manuscript.

A discussion about the different cost (without going into details) of the different scenario would be appreciated. In particular, to show how the implementation costs would affect directly the population, rather than the Municipalities or the Government.

Authors’ response:

The implementation costs are an interesting topic, but since they were not studied in this paper, the discussion on it was not extensive. We have added some text to the paragraph in the discussion concerning costs.